

# High resolution LGM climate over Europe and the Alpine region using the regional climate model WRF

Emmanuele Russo[1,2,3], Jonathan Buzan[1,2], Sebastian Lienert[1,2], Guillaume Jouvet[4,5], Patricio Velasquez Alvarez[1,2,3], Basil Davis[5], Patrick Ludwig[6], Fortunat Joos[1,2], and Christoph C. Raible[1,2]

[1]Climate and Environmental Physics, University of Bern, Sidlerstrasse 5, 3012, Bern, Switzerland.
[2]Oeschger Centre for Climate Change Research, University of Bern, Hochschulstrasse 4, 3012, Bern, Switzerland.
[3]now at: Institute for Atmospheric and Climate Science (IAC), ETH Zurich, Universitätstrasse 16, 8092 Zurich, Switzerland.
[4]Department of Geography University of Zurich, Zurich, Switzerland
[5]Institute of Earth Surface Dynamics, Geopolis, University of Lausanne, Switzerland
[6]Institute of Meteorology and Climate Research, Karlsruhe Institute of Technology, Karlsruhe, Germany

**Correspondence:** emmanuele.russo@env.ethz.ch

**Abstract.** In this study we present a series of sensitivity experiments conducted for the Last Glacial Maximum (LGM, ∼21000 years ago) over Europe using the regional climate Weather Research and Forecasting model (WRF). Using a 4 step 2-way nesting approach, we are able to reach a convection-permitting horizontal resolution over the inner part of the study area, covering Central Europe and the Alpine region. The main objective of the paper is to evaluate a model version including a

series of new developments better suitable for the simulation of paleo-glacial time slices with respect to the ones employed in former studies. The evaluation of the model is conducted against newly available pollen-based reconstructions of the LGM European climate and takes into account the effect of two main sources of model uncertainty: a different height of continental glaciers at higher latitudes of the northern hemisphere and different land cover. Model results are in good agreement with evidence from the proxies, in particular for temperatures. Importantly, the consideration of different ensemble members for

characterizing model uncertainty allows to increase the agreement of the model against the proxy reconstructions that would be obtained when considering a single model realization. The spread of the produced ensemble is relatively small for temperature, beside areas surrounding glaciers in summer. On the other hand, differences between the different ensemble members are very pronounced for precipitation, in particular in winter over areas highly affected by moisture advection from the Atlantic. This highlights the importance of the considered sources of uncertainty for the study of European LGM climate and allows to

determine where the results of an RCM are more likely to be uncertain for the considered case study. Finally, the results are also used to demonstrate the added value of convection-permitting resolutions, at both local and regional scales, under glacial conditions.

## 1 Introduction

The Last Glacial Maximum (LGM) is known as the time at which the northern hemisphere ice-sheet volume reached its

maximum during the last glacial period, at around 21000 years Before Present (BP) (Clark et al., 2009). The radiative forcing of the LGM was considerably different than the Pre-Industrial (PI) period. Atmospheric greenhouse gas (GHG) concentrations



were lower at LGM than at PI. At the same time, changes in the orbital configuration of the Earth around the Sun induced a different seasonal pattern of incoming insolation during the LGM, with lower values over the higher latitudes of the Northern Hemisphere during summer, with respect to the present-day (Berger, 1978; Loulergue et al., 2008; Kageyama et al., 2017; Bereiter et al., 2015). The direct effect of different radiative forcing led to important changes at the global scale in the climate

of the LGM compared to the present-day, with overall cooler and drier conditions. Syntheses of proxy-data and model outputs indicate that annual global temperatures were in a range of 4 to less than 8°C lower than their PI values (Osman et al., 2021; Hargreaves et al., 2011). At the same time, feedback mechanisms, such as the ice-albedo effect, land cover changes and ice sheet expansion, played an important role in modulating the climate of the LGM, not only at the global scale but also at a regional and local level. For example, a large extension of northern hemispheric ice-sheets had a strong impact on the large-

scale circulation, with a different downward impact over different parts of Europe (Hofer et al., 2012b, a; Löfverström et al., 2014; Merz et al., 2015; Löfverström and Liakka, 2016; Löfverström et al., 2016; Ludwig et al., 2016; Kageyama et al., 2017).

The study of the climate of the LGM offers the opportunity to better understand processes and feedbacks of the climate system that have no analogue for the present-day and the near-future (Kohfeld and Harrison, 2000; Kageyama et al., 2017; Raible et al., 2020). For this reason, the LGM represents a unique opportunity for evaluating the response of climate models

to an extreme change in climate forcing, improving model reliability and contributing to model development (Kohfeld and Harrison, 2000). Consequently, the LGM has been one of the main target periods of paleoclimate modeling studies, with the first attempts at reproducing and understanding LGM climate dating back more than 40 years (Alyea, 1972; Williams et al., 1973; Kutzbach and Guetter, 1986; Rind, 1987; Gates, 1976; Manabe and Hahn, 1977).

The complexity of processes and feedbacks that directly or indirectly influenced the LGM climate makes the task of re-

producing it via dynamical models particularly challenging. Over the years, this has led to models of increasing complexity being applied to the study of the LGM climate, from intermediate complexity models to fully coupled EARTH System Models (ESMs) (Braconnot et al., 2012; Harrison et al., 2014, 2015; Annan and Hargreaves, 2015; Masson-Delmotte et al., 2006; Izumi et al., 2013; Li and Morrill, 2013; Lambert et al., 2013; Schmidt et al., 2014; Otto-Bliesner et al., 2007; Muglia and Schmittner, 2021, 2015; Menviel et al., 2017; Sime et al., 2013; Kageyama et al., 2013; Hargreaves et al., 2011). However,

these increases in model complexity have not generally led to improved model performance when compared against proxy data (Harrison et al., 2014; Annan and Hargreaves, 2015).

In recent years, Regional Climate Models (RCMs) have been employed for the study of LGM climate, mainly motivated by their improved representation of local processes and having a spatial resolution better matched to those of proxy reconstructions with respect to GCMs. In particular, several studies have shown that RCMs for the LGM can improve the simulated climate

in comparison with the driving GCM simulation (Strandberg et al., 2011; Ludwig et al., 2019). Recently, a series of LGM studies have shown that areas with mountainous terrain, such as the Alps, profit from the use of an RCM with convection-permitting resolution. They have also highlighted the important role of land-surface characterization for the representation of LGM climate over Europe (Velasquez et al., 2020, 2021, 2022).

It is important to acknowledge that the added value of RCMs for paleo applications is still under debate. While some studies

highlight that an RCM signal is too much dictated by its driving GCM, playing a major role (Armstrong et al., 2019), others





consider the relatively low computational demands of RCMs as an added value compared to their driving GCMs, since this allows for a more comprehensive characterization of uncertainties related to changes in soil and surface features (Russo et al., 2022). Both points are actually very important for the LGM, for which high uncertainties in RCM simulations may derive from the so called "garbage in-garbage out" effect related to the imposed GCM signal and from differences in the characterization of

surface features such as land cover and ice height (Kjellström et al., 2010; Strandberg et al., 2011). These sources of uncertainty have to be properly taken into account when willing to assess the climate of the LGM using an RCM.

In this study, we present a series of LGM sensitivity experiments performed over Europe with the RCM WRF 3.8.1 (Skamarock et al., 2008; Powers et al., 2017). The model includes some important technical developments with respect to the version used in the former studies of Velasquez et al. (2020, 2021, 2022), better suitable for a proper application of the model

in paleo-studies. The performed experiments are built considering two different land-cover datasets, as well as changes in continental and Alpine glaciers extent for both WRF and its driving GCM. Model results are evaluated against a newly developed pollen-based reconstruction database for the European LGM climate. The main goals of the study are the evaluation of a new model version against proxy-reconstructions, the characterization of model uncertainties resulting from changes in the simulations setup relative to land cover and ice height and the assessment of the added value of convection-permitting

simulations for paleoclimate studies. At the same time, with a main focus on the Alpine climate, the study aims to produce high-resolution outputs to force a glacier model, and reconstruct the extent of glaciers during glacial maxima (Jouvet et al., under minor revision).

Section 2 gives a general overview of the different methods and data used in this study. In Section 3, the results are presented: first a comparison against the proxy-based reconstructions is conducted for summer and winter values of temperatures and

precipitation in Section 3.1; secondly, the role of different uncertainties is considered for the same seasons and variables in Section 3.2; thirdly, some considerations on the role of convection-permitting resolutions are drawn in Section 3.3. Finally, a summary of the obtained results is discussed in Section 4.

## 2   Model, data and methods

The results of this study are based on a set of simulations performed with the RCM WRF 3.8.1 (Skamarock et al., 2008; Powers

et al., 2017). The starting point of the presented simulations are the results of earlier studies using the same model version (Velasquez et al., 2020, 2021, 2022). In the following, first the driving data used to conduct the performed simulations are introduced in Section 2.1. Successively, the general model setup is presented in Section 2.2, highlighting the main differences and improvements applied for the use of WRF in paleoclimate studies with respect to the previous version of Velasquez et al. (2020, 2021, 2022). Then, the complete set of performed LGM experiments is described in detail in Section 2.3 . Finally, the

new proxy reconstructions used for the assessment of the LGM model performance are introduced in Section 2.4.



## 2.1 Driving Data

RCMs need climate information at their lateral boundaries. Here, outputs of a series of equilibrium climate simulations performed in a model chain of the Community Earth System Model (CESM 1.2 version, Hurrell et al. (2013)) are used. CESM is a state of the art Earth System Model (ESM) consisting of different model components for the description of the atmosphere,

land-surface, ocean and sea-ice. In a first step, a fully coupled simulation at a horizontal resolution of $1.9° × 2.5°$ for the atmospheric and the land components, and of nominal $1° × 1°$ for the ocean and the sea-ice components are executed (Buzan et al., 2023). Then, time varying sea-ice mask and sea surface temperatures are derived from the simulation and are prescribed, in a successive step, to another experiment using only the atmosphere and land components of the CESM over an integration time of 24 years, at a horizontal resolution of $1.25° × 0.9°$. Through these two steps, it is possible to reach a high spatial

resolution, allowing for a realistic representation of the large-scale atmospheric circulation (Merz et al., 2015), while keeping the computational costs low.

For all of the CESM LGM simulations employed in this study, values of the orbital parameters (obliquity 22.949°, eccentricity 0.018994 and precession294.425°), greenhouse gas concentrations ($CO_2$ 190 ppm, $CH_4$ 375 ppb, $N_2O$ 200 ppb) and land use changes for the corresponding period at 21000 years BP are used following the directives of the Paleoclimate Model

Intercomparison Project 4 (PMIP4, Kageyama et al. (2017)).

The reference simulation is performed using the ICE-6G ice sheet reconstruction of Peltier et al. (2015), the modification of which follows the setup for the LGM PIMP4 protocol (Zhu et al., 2017). Additionally, 3 sensitivity experiments are performed (Buzan et al., 2023), where only the ice sheet height of the main northern hemisphere ice sheets (Laurentide, Greenland , Fennoscandia) is varied respectively to 33%, 67% and 125% of their height as derived from the dataset of Peltier et al. (2015).

Additional details on the CESM model set up are presented in Buzan et al. (2023).

## 2.2 WRF general model setup

The general setup of the performed WRF simulations considers 4 2-way nested domains, with horizontal resolutions of 54, 18, 6 and 2 km, respectively (Fig. 1). The domains cover the entire European region, with a focus over the Alps. For the time integration, an adaptive time-step is used with a minimum time step of 108, 36, 12 and 4 seconds, respectively for the domains

going from coarser to higher resolutions. The simulations are conducted considering a total of 40 vertical eta levels in the atmosphere, and 4 layers in the soil, with varying vertical resolution. The Kain-Fritsch cumulus convection scheme (Kain and Fritsch, 1990, 1993; Kain, 2004) is used for the coarser domains **d01** and **d02** of Fig. 1. In the inner domains **d03** and **d04**, the convection parameterization is switched off since higher resolutions permit to get an explicit representation of convective processes. For all 4 domains, the Land Surface Model NOAH-MP (Yang et al., 2011b, a) is used. For long-wave and short-

wave radiation, the Dudhia and the RRTM schemes are respectively employed. The different LGM GHG values cannot be set in the model namelist, but are passed directly to the corresponding RRTM shortwave radiation module prior to the model compilation, consistently with the values of the CESM simulations, as specified in Table 1. A general overview of the model setup employed for the different domains is presented in Table 2.



Having introduced the main features of the WRF simulations in common with Velasquez et al. (2021), we successively describe the main differences with respect to this former study. The initial setup of Velasquez et al. (2021) included a glacier scheme of NOAH-MP, allowing ice phase changes in the soil, that was found to be incorrect, producing unrealistic soil temperatures. The here presented simulations use a slab ice scheme of NOAH-MP that does not consider ice phase changes in the soil, resulting in more realistic soil temperatures than the former option. Another major improvement, particularly relevant for the application of WRF to paleoclimate studies, concerns the model representation of changes in the orbital parameters of the Earth on millennial timescales. In fact, while different values of the obliquity of the orbit can be passed directly into the code of the default model version 3.8.1, this is not possible for the different values of the parameters representing changes in the eccentricity of the Earth's orbit around the Sun and in the precession of the equinoxes. For this reason, a FORTRAN subroutine already employed in WRF version 4.1.2 in Ludwig and Hochman (2022) is implemented in the main radiation module of the model to include the full orbital forcing of the LGM experiments. The routine allows to scale the value of the solar constant depending on the effective position of the Earth on its orbit, taking into account changes in the orbital parameters. It is developed from a subroutine used in several other paleoclimate studies with the RCM COSMO-CLM (Fallah et al., 2016, 2018; Prömmel et al., 2013; Russo and Cubasch, 2016; Russo et al., 2022), and is based on an original subroutine used in the GCM ECHAM5 (Roeckner et al., 2003), considering basic Kepler laws only and using the long-term series expansions of Laskar et al. (1993). The effect of the implemented subroutine on the seasonal pattern of insolation is here assessed for the coarsest study domain **d01**. Fig. 2 shows the differences in incoming radiation on top of the atmosphere calculated for each day of a year between the LGM and the PI periods, for the new (left) and the default (right) version of the model. The results with the new treatment of orbital parameters (2, left) are now in-line with the expected seasonal pattern of insolation at the LGM over the study domain (Kageyama et al., 2017).

## 2.3 WRF sensitivity experiments

The CESM simulations with a spatial resolution of $1.25° \times 0.9°$ are used as initial and boundary conditions to run WRF. A total of 5 simulations with convection-permitting resolutions are performed with the WRF model for the LGM, each one covering approximately 11 years. For each of these simulations, given the extremely high computational demands of very high-resolution, each period is divided into 3 sub-periods of different length, in between 52 and 40 months and considering the first 4 months as spinup time. In the end, the results of these sub-periods are joined together to retrieve a mean climatology for each of the performed experiments, based on the mean calculated over a total of 10 years. The approach is similar to the one of previous studies such as the one of Velasquez et al. (2020, 2021, 2022). It might not allow to exhaustively assess the interannual variability of a model over a given period of study, due to the short time frames considered for each experiment. Nevertheless, since the interest of this study is mainly on climatological mean values, this represents a good compromise between high demand in computational resources and the length of a signal to be used for building up a climatology.

To perform experiments with WRF under LGM conditions, further modifications to the surface boundary conditions are necessary. Changes to land cover, soil composition, ice extent, ice height and the land-sea mask are applied here to the original present-day WRF data-sets. A reference simulation, here referred to as DEF (Table 3), is performed using the outputs from



the reference CESM simulation of Buzan et al. (2023) as boundary conditions. For the DEF experiment, the LGM topography is prescribed from the dataset of Peltier et al. (2015). In particular, the maximum ice height in the period from 24000 and 18000 years Before Present (BP) as derived from Peltier et al. (2015) is considered in this case. For the Alpine region, the ice-cap elevation is derived using the Parallel Ice Sheet Model (PISM, Khroulev and the PISM Authors, 2020) forced by the

WRF LGM simulation of Velasquez et al. (2021), and tuned such that the glacial maximum area fits the reconstructed one by Ehlers et al. (2011). The land-sea mask of the DEF simulation is derived from Peltier et al. (2015), also considering the largest extension of ice in between 24000 years BP and 18000 years BP.

   Three additional simulations are conducted using different boundary data derived from the CESM simulations with perturbed LGM ice height over the higher latitudes of the Northern Hemisphere. Consistently with the driving data, for these 3 simulations

the height of the northern latitudes ice sheets of the DEF simulation over the considered domain of study is reduced by 33% and 67%, and increased by 25%. These simulations will be referred to, in the following text, respectively as 67ICE, 33ICE and 125ICE (Table 3).

   For the LGM land cover, the LGM simulation by Velasquez et al. (2021) is used to drive offline the dynamic vegetation Land surface Processes and eXchanges model of the University of Bern (LPX-Bern v1.4) (Lienert and Joos, 2018; Spahni

et al., 2013; Stocker et al., 2013). LPX-Bern features dynamic carbon, water, and nitrogen cycles and vegetation is internally represented by 10 Plant Functional Types (PFTs; 8 tree and 2 herbaceous) competing for resources and adhering to bioclimatic limits. Input variables for LPX-Bern are the fraction of land in a grid cell and monthly values of total precipitation, near-surface temperature, and net downward shortwave radiation at surface. Furthermore, for Nitrogen deposition pre-industrial values of the forcing product (Tian et al., 2018) closest to the model gridcell are used. LPX-Bern horizontal resolution is adapted to

match the horizontal resolution of the respective WRF domains. For using the LGM maps of vegetation derived from LPX-Bern into WRF, the 12 biomes of LPX-Bern need to be translated into one of the 16 classes of WRF (based on the data from the US Geological Survey (USGS)). The selected BIOME correspondence between the 2 different datasets is presented in Tab 3.

   All the 4 above introduced simulations (DEF, ICE33, ICE67 and ICE125) use the same vegetation map derived with LPX-

Bern from the simulation of Velasquez et al. (2021). Additionally, in order to acknowledge the possible effect of uncertainty in the prescribed land cover, a simulation is performed by driving offline LPX-Bern with the climatological outputs of the DEF simulation. This simulation will be referred to as the **BIOME** simulation. The DEF and the BIOME experiments are run using the same CESM boundary conditions at $1° \times 1°$ resolution. A map of the 2 different land cover data-sets used for the conducted experiments is presented in Fig. 3. For a large part of the domain there are noticeable differences in vegetation, both

on a continental and local scale. Only over a remote part of North-eastern Europe the 2 data-sets seem to correspond. For this reason, the designed experiments are likely to provide important information on the role of uncertainty related to the use of a different land cover in RCM simulations of European climate at the LGM. Present-day soil categories are used for all the LGM simulations, with some modifications: all the soil layers in correspondence of a point covered by glaciers are set to ice. Consistently, for a given grid box, the same soil category of the upper soil layer is used for all soil levels at different depths.





Finally, in addition to the already described simulations, an experiment with the same setup of the DEF simulation but using only two nested domains, D01 and D02, down to a spatial resolution of 18 km and with the convection parameterization switched on, is performed . This experiment is indicated as DEF_noconv in Table 3 and is conducted with the goal of better assessing the role of convection-permitting simulations for the representation of the European LGM climate.

**2.4   Proxy reconstructions data-set**

The model simulations are evaluated against a new pollen-based reconstruction of LGM seasonal and annual temperature and precipitation by Davis et al. (2022). This reconstruction is based on pollen data from 63 pollen records that were selected for the quality of their dating control over the LGM period (21,000 BP ±2000 Cal. BP). Records were excluded if they fell within category 7 ('poor') of the 1-7 COHMAP scale of chronological quality based on the proximity of dates to the LGM

time-slice. Only absolute dating methods (e.g. radiometric dating such as radiocarbon) were considered, and relative dates based on, for instance pollen correlation, were excluded. The record selection process, therefore, excluded many of the pollen records used in previous studies, reflecting the generally poor quality of the dating control in LGM pollen records, including 10 of the 18 the records used in the PMIP benchmarking dataset (Bartlein et al., 2011). The reconstruction itself uses a standard modern analogue technique (MAT) that finds the best 6 analogues amongst a dataset of over 8000 modern pollen samples

from the Eurasian Modern Pollen Database (Davis et al., 2020). The MAT method has been used for previous LGM pollen-climate reconstructions (Peyron et al., 1998) but it has not been known how much the method may be influenced during the LGM by problems such as no-modern-analogue vegetation (and climate), as well as changing plant physiological responses to climate associated with lower atmospheric $CO_2$. For this reason, the method was evaluated against a previous pollen-climate reconstruction based on inverse modelling (Wu et al., 2007), which is designed to account for many of these problems, as well

as a chironomid-based reconstruction (Samartin et al., 2016). The result of this evaluation showed little difference between the MAT reconstruction and these other methods, indicating that the MAT method is reliable for this time period and location. This can be attributed in part to advances in the size and spatial coverage of the modern pollen training set which is an order of magnitude larger than those used in previous MAT reconstructions, permitting good vegetation analogues (chord distance $< 0.3$, Huntley (1990)) to be found for all fossil pollen samples.

**3   Results and discussion**

**3.1   Comparison of the reference simulation against pollen-based reconstructions**

In Fig. 4 we present the mean climatologies of winter and summer temperatures and precipitation of the DEF LGM simulation over the coarsest domain of the study, together with the corresponding values derived from the pollen-based reconstructions. Winter temperatures below 0°C characterize almost the entire domain for the DEF experiment, except for a few points over

Southern Spain and Northwestern Africa. Particularly cold simulated temperatures are evident in winter over the north-eastern part of Europe, with values generally below -25°C. Temperatures remain lower than 0°C in the model in summer over North



and Northeastern Europe, over areas largely covered by glaciers. Simulated temperatures rarely exceed a mean value of 20°C in summer, only over specific areas of Southern Europe and Northern Africa. The values of the pollen-based reconstructions, displayed in Fig. 4 as filled circles, show a similar behaviour, even though their coverage is mainly limited to Central and Southern Europe. In winter, the pollen data are also characterized by a north-east to south-west gradient in temperatures, with

values consistent with the ones of the WRF DEF experiment. In summer, reconstructed temperatures are similar to the ones simulated by WRF for the entire domain, presenting in particular a similar north-to-south gradient. The simulated precipitation pattern in winter is characterized by high values in the western areas of the domain (Fig. 4, lower row), for which moisture advection from the Atlantic Ocean plays a major role (Beghin et al., 2016; Ulbrich et al., 1999), up to seasonal values of 500 mm/season in some case. In summer, drier conditions are evident over almost the entire domain, except for few areas

with complex topography such as part of the Alps and Western Scandinavia, where precipitation exceeds 300 mm/season. The pollen-based reconstructions show a different picture compared to the DEF simulation for precipitation in winter, with proxy locations in the western part of the domain not remarkably wetter than the ones from the central and eastern parts of the domain. Additionally, the range of values of winter precipitation derived from the pollen-based reconstructions is generally narrower than in the case of the DEF simulation, with lower maximum values rarely exceeding 250 mm/season. For summer

precipitation, on the other hand, there is a better agreement between the simulated data and proxy reconstructions than in winter, except for specific areas, such as around the Alps. In summer, the simulated values of precipitation are in the same range of the climate reconstructions.

## 3.2 Consideration of different model uncertainties

A more quantitative assessment of the differences in seasonal values of temperature and precipitation between the DEF WRF

simulation and the pollen-based reconstructions is provided in Fig. 5 and Fig. 6, respectively. In these figures, also the temperature and precipitation derived from the additional four WRF simulations (namely BIOME, 33ICE, 67ICE, 125ICE) are considered. The comparison is additionally conducted on both the coarse resolution domain, and the inner domain d04 of Fig. 1 in this case. A circle with the values of the differences between the reference simulation DEF and the pollen-based reconstructions is drawn for the proxy locations. If the bias of the reference simulation DEF lies within 1 standard deviation

of the pollen values, then a black dot is added at the center of a given circle. Successively, if at least one of the members of the ensemble lies within 1 standard deviation of the values of the pollen-based reconstructions, then the circle is highlighted in red. Finally, the maximum absolute differences in the considered variables calculated between the different ensemble members are plotted for each point of the domain on a gray-scale, in order to provide an assessment of the ensemble spread and model uncertainty.

Fig. 5 confirms that, for both winter and summer, there is a very good agreement between the performed simulations and the values derived from the proxies. The consideration of the different simulations with the applied changes in the model configuration allow to get very close to the evidence of the proxies for almost the entire points of the domain, in both seasons and for both considered variables. For winter temperatures, the bias of the reference (DEF) exceeds in very few cases an absolute value of 5°C. Interestingly, despite the large changes in the configuration of the different ensemble members, their





results in terms of winter temperature are relatively close for almost the entire domain of study. Differences between the different ensemble members larger than 4 °C are evident for winter temperature mainly over the north-eastern part of the domain and Iceland, with values spreading over a range of more than 10°C in these regions. For summer, again, the mean model temperatures are very close to the values of the pollen-based reconstructions over the entire domain (Fig. 5, right column). The

DEF simulation, in particular, shows a small bias against the proxies over the entire domain, with values rarely exceeding 4°C and well within the range of the pollen uncertainty. The main exceptions are some points over the area of present-day France, for which a closer agreement is reached only when considering other members of the ensemble. In summer, more remarkable and spatially heterogeneous differences in temperatures are evident between the different ensemble members than in winter. In particular, high differences of up to 14°C are found for areas at the border of glaciers over Scandinavia and Iceland.

Analyses for the inner Alpine domain (Fig. 5, bottom row) confirm an overall good match between temperatures derived from the performed simulations and the pollen-based reconstructions, for both summer and winter. In this case, the main differences between the different ensemble members are evident in winter over the Po valley, exceeding in some cases 6°C. Just 2 points of the pollen reconstructions, in summer, have temperature values outside the range of the ensemble members. This gives high confidence on the use of the presented results for the study of the area.

For precipitation (Fig. 6), the agreement between model simulations and proxy reconstructions is not as good as in the case of temperature. In winter, the model seems to be generally too wet, in particular over the Northern Iberian Peninsula, present-day France, the Alps and continental areas of present-day Greece and Turkey, with seasonal values of precipitation exceeding the ones derived from pollen-based reconstructions by more than 300 mm/season. For these areas, none of the performed simulations gets closer to the range of values of the proxies. In winter, the area for which the DEF simulation

presents higher values of precipitation, i.e. Western Europe (Fig. 4, lower row), are the same for which higher differences in the ensemble members are evident, exceeding in most of this area 150 mm/season. A similar overlapping of the largest differences in the ensemble members over regions presenting the highest values of precipitation in the DEF simulation is also evident in summer. In this case though, there is a better agreement between the results of the performed simulations and the pollen-based reconstructions, although the model seems to be generally drier, in particular over the Balkans, where the

differences against the pollen-based reconstructions are in some cases below -200 mm/season. Worth to mention in this case is that changes in land cover seem to be very important for the representation of summer precipitation over the Mediterranean region, in particular over the Balkans (Fig. S1 ̈in the Supplementary Information; SI), where the BIOME simulation produces significant changes with respect to the DEF experiment.

For the inner domain of study there is also a relatively good agreement between the proposed simulations and the pollen-

based reconstructions in terms of precipitation, for both seasons, but this is again worse than in the case of temperature (Fig. 5, lower row) at least when considering the number of points for which the simulations lie within the pollen uncertainty. Three important points need to be mentioned in the case of precipitation for the inner domain of study. Firstly, the differences between the different ensemble members are remarkable over all of the Alpine range, in both winter and summer, with values extensively exceeding 200 mm/season. This confirms again the relevance of the considered sources of uncertainty in the model

for the investigation of the LGM climate of the region. Secondly, even though the simulations are out of the range of the



considered pollen-based reconstructions over some points in one season, they get closer in the other. This indicates that some of the results, even if not perfectly matching the values of the proxies, are not completely off. Thirdly, the pollen reconstructions are trained with present day observations and studies show that even observations nowadays tend to underestimate precipitation in particular areas characterised by complex terrain (Fu, 1996; Frei and Schär, 1998; Frei et al., 2003).

## 3.3 Considerations on the role of convection-resolving resolutions

In Fig. 7 the differences in the mean seasonal values of temperature and precipitation between the reference DEF simulation and the DEF_noconv experiment are calculated for each grid box of the outer domain d01. The main goal of such an analysis is to assess whether the explicit representation of convection over the inner part of the domain has a relevant impact on the simulated variables at both the local and continental scale, with respect to a simulation exclusively using a parameterised representation of convection over all the domain. In Fig. 7, for each grid-box where the daily mean anomalies of the considered variable and season can be considered significantly different according to the results of a Kolmogorov-Smirnof (KS) significance test at a significance level of 0.05, a dot is plotted.

Results show that for the Alpine region, the effects of higher resolution and the explicit representation of convection are significant, for both variables and seasons. For precipitation, in particular, the effects of the convection-permitting simulation are seen not only over the inner domains, but also over the entire domain of study, presenting significant differences between the two simulations. These differences are in some case of the same order of the differences between the different sensitivity tests conducted by perturbing the glaciers ice height and extension (see SI, Fig. S1), and considering considerably different boundaries. Therefore, the consideration of convection-permitting resolutions for the simulation of European LGM climate has a significant impact on the results, comparable to the effect of other relevant sources of uncertainty, not only at a local scale but over the entire European domain.

## 4 Conclusions

In this study, we conduct an evaluation of a version of the RCM WRF 3.8.1, including a series of technical developments suitable for paleoclimate applications, against newly available pollen-based reconstructions for the LGM climate of Europe and the Alpine region. Through the performance of a set of sensitivity experiments taking into account the role of different large-scale and surface model error sources, we aim to assess the general performance of the model. At the same time, we quantify the possible effect of changes in the model setup on the obtained results, highlighting where results of RCMs can be considered more robust and where factors such as error in the representation of surface features could play a major role in the reconstruction of the European LGM climate.

Results show that the model ensemble produced by changing surface drivers and using different boundaries is in very good agreement with the considered pollen-based reconstructions for summer and winter values of temperature. For precipitation, the agreement is also relatively good, but in this case none of the produced ensemble members gets as close to the proxies over the entire domain of study as in the case of temperature. In particular, the model results are too wet in winter over Western



Europe and too dry in summer over the Eastern part of Europe, with respect to the pollen-based reconstructions. The model shows a north-east to south-west positive gradient of temperatures in winter over Europe at the LGM, in accordance with the picture drawn from the pollen-based reconstructions. A large part of the European domain presents mean winter values below 0°C as derived from the model simulations. A north-to-south positive gradient characterizes simulated summer temperatures,

again similarly to evidence from the proxies. In summer, mean simulated temperatures rarely exceed 20°C. The LGM WRF precipitation shows particularly high values, exceeding in some case 400 mm/season over Atlantic regions of Western Europe and the Alps. In summer, the pattern of precipitation in the model is quite homogeneously distributed, with values rarely exceeding 150 mm/season, except for specific areas characterized by complex topography, such as Scandinavia and the Alps. The consideration of different northern hemispheric ice-height and land cover types in the model leads to largely differing

precipitation values over specific areas, with values exceeding 200 mm/season in some case. This allows to increase the match between the model and the proxies, at the same time highlighting the importance of the considered sources of uncertainty for the simulation of European climate at the LGM over different regions. Errors associated to changes in the ice-height of the higher latitudes of the northern hemisphere produce very pronounced changes in both winter and summer precipitation over Western Europe and the Alpine region. Changes in land cover also play an important role for summer precipitation over

the Mediterranean basin. The considered changes have a relatively limited impact on winter temperatures over almost the entire domain, playing instead a larger role on summer temperatures, in particular over areas at the boarders of glaciers, with differences between the different ensemble members reaching 14°C in some case. Finally, the use of 4 2-way nested domains, allowing to reach a spatial resolution in the inner domain of approximately 2×2 km and to explicitly represent convection, leads to important changes with respect to an experiment with parameterised convection over the inner part of the domain.

Significant differences are evident not only over the inner Alpine region, but also over the outer domain of study. These changes are particularly pronounced for precipitation rather than for temperature, being in the range of the changes relative to the use of different ice-height and land-cover.

The paper sets the basis for an improved representation of the LGM climate of Europe. It introduces some important technical developments useful for the application of WRF to the study of the LGM with respect to previously employed model versions,

allowing to obtain seasonal values of temperature and precipitation in very good agreement with newly available pollen-based reconstructions. Additionally, it permits to assess the role of different sources of model uncertainty, as well as where and for which variable largest differences in model results might be expected as a result of a different model setup. Finally, the study allows to assess the added value of simulations with convection-permitting resolutions for the study of the LGM climate.

*Code and data availability.*   WRF is a community model that can be downloaded from the following web page: (**?**, https://www2.mmm.ucar.

edu/wrf/users/, last access 04 April 2022) The climate simulations employed in this study (global: CESM and regional: WRF), as well as the land cover simulations, occupy several terabytes of memory and thus are not freely available. Nevertheless, they can be accessed upon request to the contributing authors. The scripts used for the presented analyses are available at: doi.org/10.5281/zenodo.7998789.



*Author contributions.* ER, JB and CCR contributed to the design of the experiments. ER carried out the regional simulations and wrote the first draft. JB carried out the global simulations. SL performed the LPX simulations and PV provided first input data. BD provided the proxy reconstructions. All authors contributed to the interpretation of the results, the writing, and scientific discussion.

*Competing interests.* The authors declare no competing interests.

5 *Acknowledgements.* This work was supported by the Swiss National Science Foundation (SNF) within the project "Modelling the ice flow in the western Alps during the last glacial cycle" (grant no. 200021-162444) and the NAGRA project HicAp. The simulations are performed on the supercomputing architecture of the Swiss National Supercomputing Centre (CSCS).



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



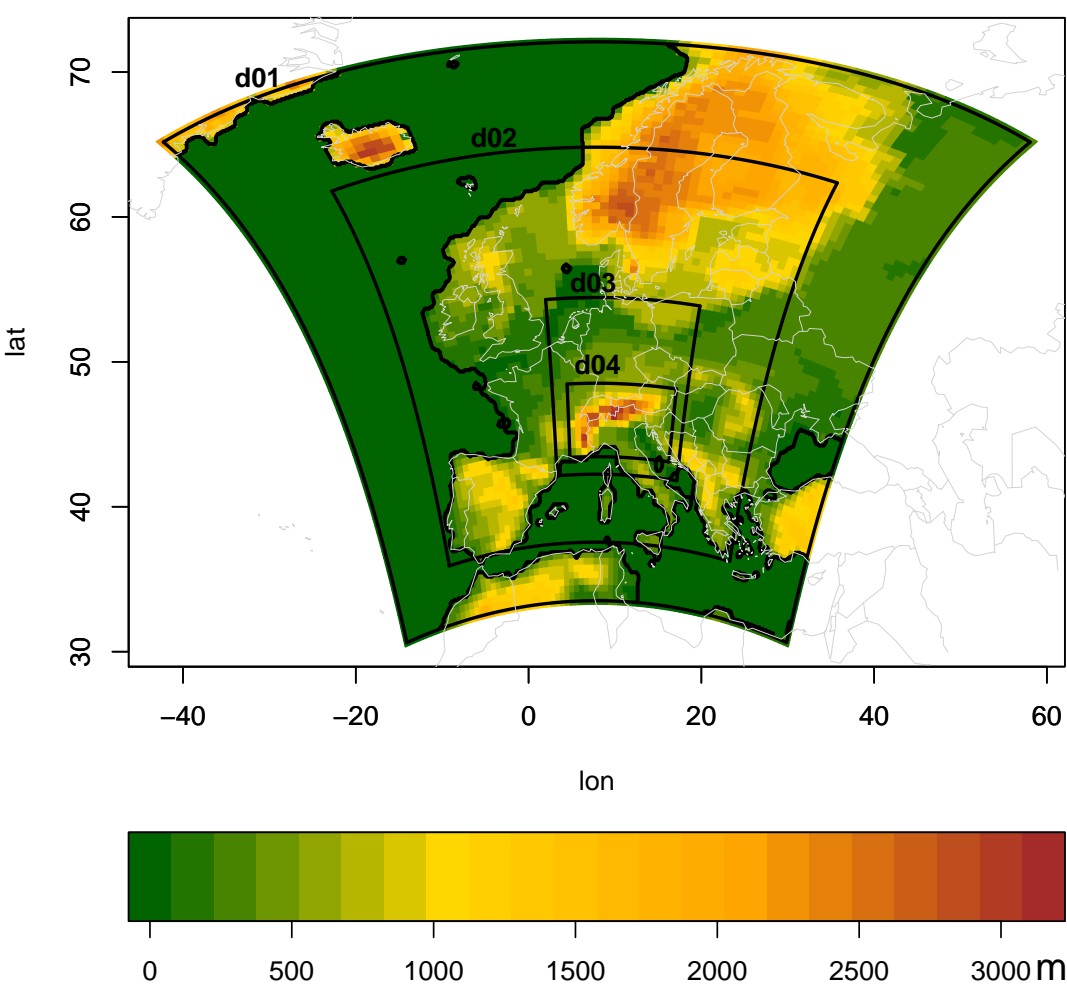

**Figure 1.** Maps of the topography and the different nested simulation domains.



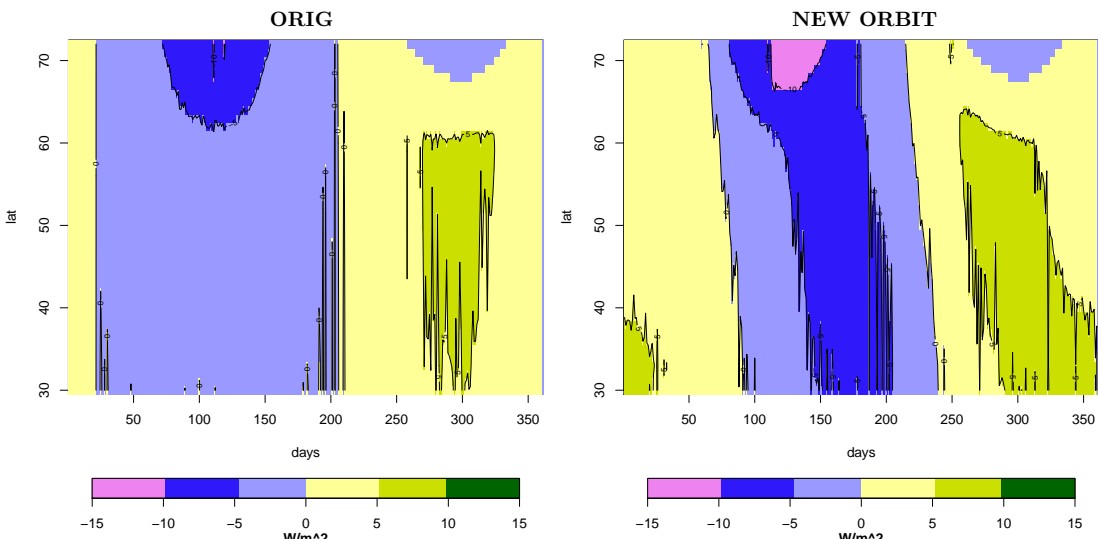

**Figure 2.** Zonal changes in daily solar insolation over the days of a year, calculated between the LGM period (∼ 21000 years BP) and present days, as in the default WRF version (ORIG, *left*) and with the newly introduced orbital routine (NEW ORBIT, *right*).





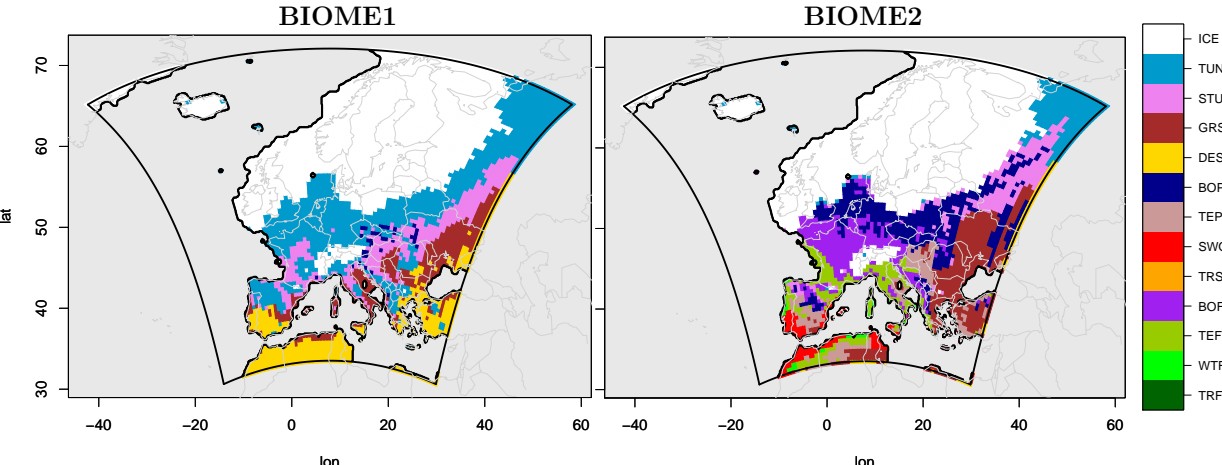

**Figure 3.** LGM land cover and biome classes for the coarser domain of study d01, as derived using the vegetation model LPX-Bern using as input the results of the WRF LGM simulation of Velasquez et al. 2021 (*left*, BIOME1) and using as input the results of our new DEF simulation (*right*, BIOME2). Beside land ice cover indicated as ICE, all the legend acronyms of the different biome species derived from LPX-Bern are specified in Table 4.





**Figure 4.** LGM winter (*left*) and summer (*right*) climatological values of mean near surface temperature (*upper row*) and precipitation (*bottom row*) as derived from the reference WRF LGM simulation, for the domain d01 of Fig. 1, at coarser resolution. The circles represent the values as derived from pollen-based reconstructions.



**Figure 5.** Maps for the bias of winter (*left*) and summer (*right*) temperatures calculated between the WRF reference simulation and the pollen-based reconstructions. Upper row shows the results for the coarser resolution domain d01 while the bottom row presents the same analysis for the higher resolution domain d04. The colored circles show the biases calculated between the two data-sets. In the case of the bias being smaller than one standard deviation of the reconstructions, a black dot is added at the center of the circle. Considering then the 5-member ensemble, in the case of the bias of at least one of these members against the reconstructions, being smaller than one standard deviation of the latter, then the circle is highlighted in red. Finally, the gray scale represents the maximum differences for each point of the domains, between the different members of the ensemble.



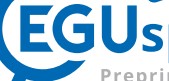

**Figure 6.** Maps for the bias of winter (*left*) and summer (*right*) seasonal precipitation calculated between the WRF reference simulation and the pollen-based reconstructions. Upper row shows the results for the coarser resolution domain d01 while the bottom row presents the same analysis for the higher resolution domain d04. The colored circles show the biases calculated between the two data-sets. In the case of the bias being smaller than one standard deviation of the reconstructions, a black dot is added at the center of the circle. Considering then the 5-member ensemble, in the case of the bias of at least one of these members against the reconstructions, being smaller than one standard deviation of the latter, then the circle is highlighted in red. Finally, the gray scale represents the maximum differences for each point of the domains, between the different members of the ensemble.







**Figure 7.** Bias in winter (*left*) and summer (*right*) values of temperatures (*upper row*) and precipitation (*bottom row*) calculated for the coarser resolution domain d01, between the reference simulation and the one with only 2 nested domains, down to 18 km. The dots in each of the figures represent the points for which the values differ significantly, at a significance level of 0.05, according to the results of a KS significance test.



**Table 1.** Orbital Parameters and Greenhouse Gases Concentration at PI and LGM periods

|       | PI        | MH        |
| ----- | --------- | --------- |
| **N2O** | 270 ppb   | 200 ppb   |
| **CO2** | 280 ppm   | 190 ppm   |
| **CH4** | 760       | 375 ppb   |
| **ECC** | 0.016724  | 0.018994  |
| **OBL** | 23.446°   | 22.949°   |
| **PRE** | 282.04°   | 294.92 °  |



**Table 2.** General description of the WRF model setup of conducted LGM simulations

|                              | d01                   | d02           | d03           | d04           |
| ---------------------------- | --------------------- | ------------- | ------------- | ------------- |
| **Spatial Resolution**       | 52km                  | 18 km         | 6 km          | 2 km          |
| **Domain Extension**         | ∼4400 km              | ∼3000 km      | ∼1200 km      | ∼1000 km      |
| **Minimum Timestep**         | 108 s                 | 36 s          | 12 s          | 4 s           |
| **Convection**               | Kain-Fritsch          | same as **d01** | NONE        | same as **d03** |
| **Time Integration**         | Runge-Kutta           | same as **d01** | same as **d01** | same as **d01** |
| **LW Radiation**             | RRTM                  | same as **d01** | same as **d01** | same as **d01** |
| **SW Radiation**             | Dudhia                | same as **d01** | same as **d01** | same as **d01** |
| **Mycrophysics**             | WSM 6-class           | same as **d01** | same as **d01** | same as **d01** |
| **Surface Layer**            | MM5 Monin-Obukhov     | same as **d01** | same as **d01** | same as **d01** |
| **Land-Surface**             | Noah-MP               | same as **d01** | same as **d01** | same as **d01** |
| **Planetary Boundary Layer** | YSU                   | same as **d01** | same as **d01** | same as **d01** |
| **Runoff & Groundwater**     | BATS                  | same as **d01** | same as **d01** | same as **d01** |
| **Surface evaporation resistance** | Sakaguchi & Zeng 2009 | same as **d01** | same as **d01** | same as **d01** |



**Table 3.** Description of conducted LGM sensitivity experiments with WRF

| Experiment Name | Description |
|---|---|
| **DEF** | Reference Simulation with setup of Table 1 |
| **ICE33** | Same setup of **DEF** but ice cap height reduced by 67% |
| **ICE67** | Same setup of **DEF** but ice cap height reduced by 33% |
| **ICE125** | Same setup of **DEF** but ice cap height increased by 25% |
| **BIOME** | Same setup of **DEF** but with land cover map derived running LPX-Bern with outputs of **DEF** |
| **DEF_noconv** | Same setup of **DEF** but only for the first two domains of Fig. 1, down to an horizontal resolution of 18 km |



**Table 4.** List of Biomes from LPX-Bern (*left*) and the corresponding ones from MODIS, they are converted to (*right*)

| LPX | MODIS |
|---|---|
| Tropical Forest (TRF) | Evergreen Broadleaf Forest |
| Warm Temperate Forest (WTF) | Mixed Forest |
| Temperate Forest (TEF) | Deciduous Broadleaf Forest |
| Boreal Forest (BOF) | Deciduous Needleleaf Forest |
| Tropical Savanna (TRS) | Woody Savanna |
| Sclerophyll Woodland (SWC) | Closed Shrubland |
| Temperate Parkland (TEP) | Evergreen Needleleaf Forest |
| Boreal Parkland (BOP) | Wooded Tundra |
| Desert (DES) | Barren |
| Dry Grassland/Shrubland (GRS) | Grassland |
| Shrub Tundra (STU) | Mixed Tundra |
| Tundra (TUN) | Barren Tundra |