# Peer review of "High resolution LGM climate of Europe and the Alpine region using the regional climate model WRF"

_EGUsphere, 2023_

## Author Comment (AC1)

Reply to
**1st Reviewer**

Dear referee,

thank you very much for accepting to review our manuscript and for the time you dedicated to its revision.

Below we go point by point through your technical corrections, presented in *italic*, detailing how we dealt with your concerns reported in **bold**.

Sincerely,

Emmanuele Russo on behalf of the author team

**Main Comments**

*Recommendation: The manuscript is, in my opinion, well and clearly written, but the motivation of the study does not come across as clearly. I have some recommendations that the authors may want to consider in a revised version*

- *The research question and motivations of the study are somewhat unclear. The study presents a new set of simulations, but why are they necessary? What were the deficiencies of previous simulations? Why is a convection-permitting resolution in principle necessary to better simulate the climate of the LGM? One of the main conclusions is that the resolving convection does have a clear effect on the simulated precipitation, but it remains unclear - unless I missed it in the manuscript - whether the convection scheme improves the simulation of precipitation (and maybe temperature) compared to the reconstructions.*

**We agree with the reviewer that the motivations of the study are somewhat unclear in the previous version of the manuscript. The paper aims at evaluating a set of new simulations of the LGM climate over Europe and the Alpine region, using an updated version of the WRF model 3.8.1 employed in previous studies. The**

updated model version includes some important bug corrections relative to the representation of ice processes in the soil as well as the introduction of a new orbital parameters routine, both of significance importance for the study of glacial times. From this point of view, the outcomes are relevant for updating previous results and for people aiming to perform new simulations for glacial periods with the same model version. At the same time, the paper considers a series of different uncertainties in the model experimental setup, showing where the results of an RCM are likely to be more uncertain for the considered case study, and for which variable, also relevant for future studies. Additionally, the produced high-resolution model outputs were initially designed to be used for simulating glaciers over the Alpine region at the LGM (Jouvet et al. [2023]). We will make all these points clearer in the new version of the manuscript. As for the role of explicitly solving convection on the simulation of the LGM European climate, in the paper we show that this could have important implications on model results, with an effect on both temperature and precipitation comparable in some case to the one of other changes in the model setup, such as modified continental ice height. This is in our opinion a very important outcome of the paper and we will try to better highlight this point in the new version of the manuscript. We also want to emphasize here that the goal of our analysis in this respect is to assess the role of convection-permitting resolutions, comparing it with other sources of uncertainty, rather than quantifying their added value for paleoclimate studies. In fact, we think that the small number of available proxy records and the relatively large size of their uncertainties would make it difficult to assess the added value of convection-permitting simulations for this case study with any statistical significance. While we do not plan to conduct additional analyses in this respect, acknowledging the referee's comment, we will expand the text on this subject to provide a more comprehensive discussion in the new version of the manuscript.

- *The model set-up also includes a small ensemble with different initial conditions (?). The study finds that the ensemble spread can be large for precipitation, but not so much for temperature. However, the length of the simulations is short, just 11 years. Is it possible that*

*the ensemble spread is just due to the short length of the simulations? Could this spread be compared to the decadal variability of precipitation in the present climate?*

Following the referee's comment, we realised that the current description of the proposed experiments is not very clear and we propose to modify it accordingly in the new version of the manuscript. The ensemble is actually not generated with different initial conditions, but with different boundaries and surface forcing. We will better specify this in the new version of the manuscript. We additionally understand the concern of the referee about the short length of each simulation for the calculation of climatological values of both temperature and precipitation. For this reason, also following a comment from the 2nd referee, we calculate here as an example the maximum range of differences in JJA temperatures between 20 different 10-year long periods derived from the 31-year long LGM simulation of Velasquez et al. 2021. Fig. 1 shows that the ensemble spread in this case is rarely exceeding 2K, compared to maximum differences of up to 14K obtained in the case of the 5-member ensemble with different boundaries presented in our paper (Fig. 5 and Fig. 6 of the former manuscript version). This suggests that even though some differences between the different ensemble members with changes in the model setup may originate from the consideration of a relatively short period of 10-year used for calculating climatological values, the presented results suggest that the large ensemble spread is not attributable to the short length of the simulations. We will take care to properly discuss this point as well as adding the new figure in the new version of the manuscript.

**Specific Comments**

*'However, these increases in model complexity have not generally led to improved model performance when compared against proxy ...'. Could the authors be more specific here? What are the deficiencies of previous simulations that remain unexplained?*

Here, we wanted to refer to the results of Kageyama et al. [2021]: "Therefore, although there are differences in the average behaviour

[Figure]

Figure 1: Maximum range of differences between the climatological values of JJA 2-meter temperature derived from 20 10-year long periods sampled from the 31-year long simulation of Velasquez et al. 2021.

across the two ensembles, the new simulation results are not fundamentally different from the PMIP3- CMIP5 results. Evaluation of large-scale climate features, such as land–sea contrast and polar amplification, confirms that the models capture these well and within the uncertainty of the paleoclimate reconstructions. Nevertheless, regional climate changes are less well simulated: the models underestimate extratropical cooling, particularly in winter, and precipitation changes." Following the referee's comment, we will try to better highlight model deficiencies that still remain unexplained according to the reported reference of Kageyama et al. [2021] in the new version of the manuscript.

- *' Model results are evaluated against a newly developed pollen-based reconstruction database for the European LGM climate.*

   *A reference to the new reconstructions would be helpful here.*

Thank you. We agree and we will add here the reference to the new reconstruction data set in the new version of the manuscript, as suggested by the referee.

- *The starting point of the presented simulations is the results of earlier studies using the same model version (Velasquez et al., 2020, 2021, 2022).'*

   *Same model version? The sentence a bit later in the paragraph says 'previous version'. Could the specific model version used by Velasquez et al. be mentioned here?*

Yes, we used the same model version of Velasquez et al. 2020, 2021, 2022. Following the referee's comment, we will revise this part of the introduction accordingly in the new version of the manuscript, also adding the specific model version used by Velasquez et al. 2020 (WRF 3.8.1) whenever necessary.

- *' D01 and D02, down to a spatial resolution of 18 km and with the convection parameterization switched on, is performed. This experiment is indicated as DEF_noconv in Table 3'*

   *switched off, I guess.*

Following the referee's comment we have realised that the description of the experiment DEF_noconv in the former version of the manuscript is not clear. In the experiment DEF_noconv convection is actually parameterised and not explicitly solved. In this case, the convection scheme is switched on. We will try to make this clearer in the new version of the manuscript, modifying the description of the experiment DEF_noconv accordingly.

- *These differences are in some cases of the same order of the differences between the different ..'*

  *...order of magnitude as the differences between.*

Thanks. We agree with the referee and we will modify the highlighted text according to the referee's suggestion.

**References**

Guillaume Jouvet, Denis Cohen, Emmanuele Russo, Jonathan Buzan, Christoph Raible, Urs Hischer, Wilfried Haeberli, Sarah Kamleitner, Susan Ivy-Ochs, Michael Imhof, Jens Becker, and Angela Landgraf. Coupled climate-glacier modelling of the last glaciation in the alps. *Journal of Glaciology*, 2023.

M. Kageyama, S.P. Harrison, M.L. Kapsch, M. Lofverstrom, J.M. Lora, U. Mikolajewicz, S. Sherriff-Tadano, T. Vadsaria, A. Abe-Ouchi, N. Bouttes, Chandan D., L.J. Gregoire, R.F. Ivanovic, K. Izumi, A.N. LeGrande, F. Lhardy, G. Lohmann, P.A. Morozova, R. Ohgaito, A. Paul, W.R. Peltier, C.J. Poulsen, A. Quiquet, D.M. Roche, X. Shi, J.E. Tierney, P.J. Valdes, E. Volodin, and J. Zhu. The pmip4 last glacial maximum experiments: preliminary results and comparison with the pmip3 simulations. *Climate of the Past*, 17(3):1065–1089, 2021.

---

## Author Comment (AC2)

Reply to
**2nd Reviewer**

Dear referee,

thank you very much for accepting to review our manuscript and for the time you dedicated to its revision.

Below we go point by point through your technical corrections, presented in *italic*, detailing how we dealt with your concerns reported in **bold**.

Sincerely,

Emmanuele Russo on behalf of the author team.

**Main Comments**

*The paper is clearly structured, and fairly well written, albeit somewhat dry, dispassionate and with little curiosity for meteorological phenomena underlying the high-resolution weather simulated, and restricting itself to the bare minimum in the climatological analysis in this version of the paper. At 4s/2km resolution, investigating for example storms, or regional effects, could be quite interesting. Some further points are detailed below. All in all a paper that can be improved.*

- *Goals: The authors aim to evaluate an apparently bug-fixed version of WRF against data, and contrast model uncertainties from ice height, land cover and convective parameterizations. These are great points to study. However, the study design does not really allow to understand how the different uncertainties play out against one another (nonfactorial), and looking at the figures in the results and discussion section, as well as the supplement, does not elucidate this further.*

**Thank you for your comment. Also following a comment from the first reviewer, we decided to review the introduction of the paper in order to make its motivations clearer. We want to clarify**

here that among the given paper goals, we do not exactly want to contrast different uncertainties, since this might be challenging given our experimental design for which change in the forcings are applied most of the times both on the RCM as well on the driving GCM. Rather, our goal is to characterize model uncertainties resulting from changes in the simulations setup relative to land cover and ice height. More specifically: "taking into account the role of different large-scale and surface model error sources, we aim to assess the general performance of the model. At the same time, we quantify the possible effect of changes in the model setup on the obtained results, highlighting where results of RCMs can be considered more robust and where factors such as error in the representation of surface features could play a major role in the reconstruction of the European LGM climate". We will try to make this point clearer in the new version of the manuscript. At the same time, following the suggestion of the referee and acknowledging the fact that it might still be important to compare different uncertainties in the model setup one-by-one, we will consider whether to expand the current figures in the supplements, including also the figures with the deviation from the reference run for the experiments considering different sources of uncertainties, for both summer and winter precipitation as well as temperature. Eventually, we will also better discuss the contents of this figure in the manuscript.

- *CO2: One potential reason why the model simulations appear biased dry is not discussed: Namely that the used pollen data suggests dryer conditions than warrented. The water-use efficiency under low CO2 conditions is lower, implying that plants are more stressed under similar climatic conditions [1] – so perhaps the model is less biased than it appears.*

First, we would like to highlight that the model is generally not always biased dry. This depends on the season and considered region. In fact, while the model results are too dry in summer over the Eastern part of Europe, with respect to the pollen-based reconstructions, they are too wet in winter over Western Europe and the Alps. In some cases, some of the considered model uncertainties help to get the model closer to the pollen reconstructed

values. This is for example the case of summer precipitation over the Eastern Mediterranean, where the consideration of different land-cover helps to reduce the model bias against the pollen. Secondly, we would like to highlight here that in our analysis we have taken into account the uncertainty of the pollen-based reconstruction data set when comparing it against model results. In particular, in line with what the referee suggests, from Fig. 6 of the former version of the manuscript it is possible to see that, for both the points characterised by a dry or wet bias, the consideration of the pollen uncertainties is very relevant: most of the model data lie within the pollen uncertainties (large amount of red circles). Also, we want to emphasize that the wet model bias in winter does not improve over just a few points over mountainous regions and for some points at the boarder of glaciers. As we have already specified in the former version of the manuscript, for areas with complex topography even for the present-day observations tend to underestimate precipitation. Therefore, in this case, this is likely not an issue related to the $CO_2$ sensitivity of the pollen data. We will try to make this point clearer in the new version of the manuscript. However, also acknowledging the possible importance of the point suggested by the reviewer we will eventually consider to briefly discuss the $CO_2$ issue of the pollen data in the new version of the manuscript.

- *Discussion: Here you could bring in more depth. You could elaborate whether you expect that the results found here dependent on the version of WRF, and on CESM as a host model? The fact that the 28-year global simulation providing input does lead to significant spread in the regional model results is surprising: Where does this divergence come from? Are these nonstationary effects that suggest that the simulation period is too short? This would also imply that averaging over such a short time period may be inappropriate, weakening the justification of one of the assumptions set out (p5 last paragraph).*

Following the comments of both reviewers we will revisit the methods and discussion sections in order to address possible uncertainties related to the fact that the model climatology computed over a period of 10 years might not be robust enough and results might change when considering different periods of time.

To prove that the differences in the different ensemble members of Fig. 5 and Fig. 6 of the former manuscript version are not the result of simply considering a too short period of time to calculate a climatology, we provide here an example of the maximum range of differences in 2-meter temperature calculated in summer from 20 10-year long periods derived from the 31-year long simulation of Velasquez et al. 2021. The largest differences between these different sub-periods very rarely exceed 2K, against a maximum value of the differences of 14K evinced when comparing the different ensemble members for the same variable. Hence, we conclude that even though the calculation of the climatology based on a 10-year period might have an effect on the given results, the applied changes in the model boundaries can be considered robust and are more important for the calculation of climatological values than the model internal variability. We will add such information in the new version of the manuscript. We will also include Fig. 1 of the current document in the supplementary material section of the new version of the manuscript.

- *Vegetation cover discussion: Given the substantial differences between the land surface conditions fed into the high-resolution simulations – don't you expect to see effects arising simply from the strongly different land cover, for example in North Africa?*

Following the referee's comment we realised that we have not properly discussed the role of land cover changes in the former version of the manuscript. These become particularly important especially over Southern Europe in summer for precipitation (See Fig. ?? in the supplements). We will try to make this point clearer in the new version of the manuscript.

**Specific Comments**

- *p2l30 "a series of LGM studies have shown..." this sentence needs references.*

Thank you. The references for this sentence are already provided one line below: "Recently, a series of LGM studies have shown

[Figure]

Figure 1: Maximum range of differences between the climatological values of JJA 2-meter temperature derived from 20 10-year long periods from the 31-year long simulation of Velasquez et al. 2021.

... profit from the use of an RCM with convection-permitting resolution. They have also highlighted the important role of land-surface characterization for the representation of LGM climate over Europe [Velasquez et al., 2020, 2021, 2022]." Following the referee's comment we will move the references at the end of the first sentence on p2l30.

- *p2l35/p3l1-3 Here a differentiation to statistical/statistical-dynamical downscaling should be added.*

Thanks. Here, we do not actually agree with the suggestion of the referee. The discussion at the highlighted lines is only inherent to RCMs. We do not think a differentiation between statistical and dynamical downscaling is required here.

- *p3l25 "The starting point .... are the results of earlier studies using the same model version"... so what? What are the results of the earlier studies that imply one should do the same things? It feels like something is missing here.*

Following the comment by the referee we have realised that this part is not very clear and we propose to modify it accordingly in the new version of the manuscript. In particular, even though the starting point of our study is the model version of Velasquez et al. 2021, we will try to make it clear in the new version of the manuscript that their model version required some important modifications for the study of the LGM, not considered before. In particular, we found that the model version used in the work of Velasquez contained a bug in the representation of ice in the soil, particularly relevant for future studies of glacial states employing the same model version.

- *p3l29 delete space after 2.3*

Thanks. We will correct the text accordingly.

- *p4l13 add space after precession*

We will correct the text according to the referee's comment.

- *p4l4 these sentences on the glacier scheme are confusing. Does ice become supercritical in NOAH-MP? Or is what is meant that there are melt/refreeze processes in the version used in Velasquez et al. (2021) that produce unphysical temperatures?*

The problem in the study of Velasquez et al. 2021 is, as highlighted by the referee in his second comment, that there are melt/refreeze processes in the soil in the model version they used, producing unphysical temperatures. Following the referee's comment we will revise this part of the manuscript, making its contents clearer.

- *p7 sec 2.4 – A key weakness of Davis et al. (2022) is that it does not address the CO2-caused precipitation bias in the reconstructions, which would be expected to cause a dry bias under the low CO2 conditions.*

As already stated in our response to the second comment by the referee, in our comparison we actually take into account the uncertainty derived from the data of Davis et al. 2022. Also, the model is drier than the pollen mainly in summer, over the Eastern Mediterranean. On the contrary, in winter the model results too wet. Again, for most of the domain, both the consideration of model and pollen uncertainties helps to bring the model closer to the pollen.

- *p8 l13-15 The narrow distribution of precipitation estimates out of the pollen-based reconstructions is perhaps indicative of the dry bias (s. above)*

The referred narrow distribution of precipitation estimates out of the pollen-based reconstructions is mainly due to the fact that the pollen have too low maxima compared to the model. We want to remark again here that in our analysis we already consider the uncertainty of the provided pollen reconstructions. However, for certain areas, even when considering such large uncertainties, the model is still far-away from the reconstructions. This is mainly

true for mountainous regions, for which we have already specified that the issue is even found for the present-day when using observational data. Following the referee's comment, we will try to make this point clearer in the new version of the manuscript.

- *p9 l27 remove " " "*

Thanks. We will correct the text accordingly.

- *p9 l32-35 Indeed, the large differences between the ensemble members are remarkable. But going back to the ensemble description, can this be simply due to internal variability in the non-overlapping subsections of the 28-year simulations? (The description of the ensemble design is confusing).*

As specified above, we conducted a test by considering 20 10-year long periods derived from the simulation of Velasquez et al. 2021. For each of these periods we calculated JJA 2-meter temperature climatological values. The maximum range of differences rarely exceeds 2K in this case, against values of 14K obtained when considering the different ensemble members with different experimental setups presented in the paper. This clearly suggests that the effect of calculating a climatology from a short period of 10-year is not very relevant with respect to the effect of the tested changes in the model setup (see Fig. 1 of the current document). We will add this information in the new version of the manuscript, also providing Fig. 1 of the current document in the supplementary material.

- *p11 Code and data availability: Fix broken reference.*

Thanks, we will correct the previously broken reference in the new version of the manuscript, as suggested by the referee.

**References**

P. Velasquez, J.O. Kaplan, M. Messmer, P. Ludwig, and C.C. Raible. The role of land cover in the climate of glacial europe. *Climate of the Past*, 17 (3):1161–1180, 2021.

P. Velasquez, M. Messmer, and C.C. Raible. The role of ice-sheet topography in the alpine hydro-climate at glacial times. *Climate of the Past*, 18:1579–1600, 2022.

Patricio Velasquez, Martina Messmer, and Christoph C. Raible. A new bias-correction method for precipitation over complex terrain suitable for different climate states: A case study using WRF (version 3.8.1). *Geoscientific Model Development*, 13(10):5007–5027, October 2020. ISSN 1991-959X. doi: 10.5194/gmd-13-5007-2020.

---

## Author Response (AR2)

Reply to
**Editor**

*Russo, E., Buzan.J, Lienert, S., Jouvet, G., Velasquez, P., Davis, B., Ludwig, P., Joos, F., Raible, C.C.: **High resolution LGM climate over Europe and the Alpine region using the regional climate model WRF***

Dear editor,

thank you very much for your latest comments on our new version of the manuscript and for the time you dedicated to its review.

Below we go point by point through your technical corrections, presented in *italic*, detailing how we dealt with your concerns reported in **Bold**.

Sincerely (on behalf of all the authors),

Emmanuele Russo

- *I*t is not very clear from the text that the DEF, xICE and BIOME experiments are part of the 5-member ensemble. Given the significant impact of the ice-sheet height on the European climate (as shown in Fig. S1-S4), does it make sense to include all these experiments in an ensemble? If you still want to proceed this way, I think that i) you should clearly explain your rationale in putting together experiments with different ice-sheet/vegetation forcing and ii) should also show and discuss the results of the different experiments alongside the "ensemble" in the main manuscript. It is indeed a pity to have performed all these experiments with different ice-sheet height and not discuss the broad impact it has on European climate.

  **Thanks for your comment. We actually agree with the editor that in the text it still results not clear that the DEF, xICE and BIOME experiments are part of the same 5-member ensemble. In fact, we have realised that we currently talk about the ensemble only starting from the results section. Following the editor's comment, we have now made it clear already in the methods section**

that the sensitivity experiments make part of a 5-member ensemble. The rationale for putting together the experiments with different boundary data and forcing into an ensemble is that we want to use them to quantitatively evaluate model results against the pollen-based reconstructions. In this context, we use the different sensitivity experiments to provide a measure of model uncertainty (here referred to as simply the range of possible model outputs obtained using the same model, but applying changes in its boundary conditions and forcing) when running an RCM for this case study. For this reason, in line with the main objectives of the paper, we still find it very appropriate to include all the experiments in the same ensemble when comparing model results against reconstructions. However, following the referee's comment, we have realised that the reasons for this choice are not clearly stated in the current version of the manuscript. Consequently, we have now specified in the methods section of the paper that the different sensitivity experiments are joined together in an ensemble in order to conduct a more quantitative evaluation of WRF against the pollen-based reconstructions, taking into account different model uncertainties. Additionally, also connected to one of your following questions, we have now specified in the methods section that by model uncertainty we refer to the range of outputs obtained with the same model, but applying changes to the model chain setup (i.e. boundaries + forcing in the case of an RCM) inherent to land cover and ice height. Concerning the comment on showing and discussing the results of the different experiments alongside the ensemble, discussing in particular the impact of changes in ice sheet height on the European climate, we do not think that this represents a research question that can be really tackled given the current design of the presented experiments. In fact, in our study the ice-sheet changes are applied each time consistently both on the RCM as well as on the driving GCM. This does not really allow us to exhaustively discriminate between the role of changes in continental ice-sheet height and changes in the large-scale atmospheric circulation generated in the driving GCM. Basically, with the performed experiments we cannot really determine whether the different model responses for the experiments with different ice-sheet height are simply due to different imposed forcing in the RCM or to changes in the boundary conditions. For this reason, as already specified in one of the answers to the reviewers, in the

manuscript we have carefully specified that we use the 5-member ensemble to provide a measure of model uncertainties resulting from changes in the simulations setup relative to land cover and ice height rather than for discriminating the role of single changes in ice-sheet height. More specifically, when introducing the paper's main objective we have stated that: "taking into account the role of different large-scale and surface model error sources, we aim to assess the general performance of the model. At the same time, we quantify the possible effect of changes in the model setup on the obtained results, highlighting where results of RCMs can be considered more robust and where factors such as error in the representation of surface features could play a major role in the reconstruction of the European LGM climate". In conclusion, for the given reasons, and also considering that with the figures currently presented in the main manuscript we are able to exhaustively answer all the 3 main proposed research questions, we have finally decided to keep the figures for the single sensitivity experiments in the supplementary part of the manuscript.

- *I*s there a similar WRF experiment for pre-industrial or present-day conditions that could be used to compare your LGM results with? Comparing simulated LGM anomalies with anomalies estimated by proxy records would provide more information on the LGM climatic change and the processes leading to them.

Thanks for your suggestion. We find this point very interesting in order to possibly shed more light onto the drivers of changes in LGM climate with respect to present-day conditions. However, currently we do not have enough resources for tackling this point, since for performing such a comparison we would need to run a present-day simulation consistent with the present-day observational dataset used for calculating anomalies in the considered pollen-based reconstructions. Consequently, we could only be able to consider this point in a possible future work.

- *P*lease show the ice-sheet forcing for the DEF and xICE experiments either in the main manuscript or SI.

Thanks. We have now included a figure with the difference in

**topography resulting from the changes in ice-sheet height of the different xICE experiments with respect to the default simulation in the main manuscript.**

- *P*lease move Fig. 2 to the supplementary as it does not seem to be central to the manuscript, and you do not compare your results to the ones obtained with experiments performed with present-day orbital parameters.

**Thanks for your comment. We have followed the editor's suggestion and included previous Fig.2 in the supplementary section of the manuscript.**

- *I*n the methods you discuss the experiment "BIOME" whereas in figure 3 you present BIOME1 and BIOME2. Please adjust the methods, so that BIOME1 and BIOME2 are accurately introduced.

**Thanks. We have now changed the subtitles of each panel of Fig. 3 following the referee's comment.**

- *Conclusions, L. 23 (tracked changes): I think you are referring here to "uncertainties related to the LGM ice-sheet height and extent as well as LGM biome distribution" and not "model uncertainties". Please make sure this is also clear in other parts of the manuscript.*

**Thanks. Following the editor's comment, as already specified in one of our previous answers above, we have now made clear in the methods section of the main manuscript that by model uncertainty we refer to the range of outputs obtained with the same model when applying changes to the model chain setup (i.e. boundaries + forcing in the case of an RCM) inherent to land cover and ice height.**

- *Figure 7: I think here you are showing "anomalies" and not "bias".*

**Thanks for pointing this out. Actually we think that the best option would simply be to call this differences between the two simulations. We have now modified the manuscript accordingly.**